# Unraveling the Contribution of Estrobolome Alterations to Endometriosis Pathogenesis

**DOI:** 10.3390/cimb47070502

**Published:** 2025-07-01

**Authors:** Giulia Nannini, Francesco Cei, Amedeo Amedei

**Affiliations:** 1Department of Experimental and Clinical Medicine, University of Florence, 50139 Florence, Italy; f.cei1@student.unisi.it (F.C.); amedeo.amedei@unifi.it (A.A.); 2Network of Immunity in Infection, Malignancy and Autoimmunity (NIIMA), Universal Scientific Education and Research Network (USERN), 50139 Florence, Italy

**Keywords:** endometriosis, estrobolome, gut microbiota, vaginal microbiota, estrogen, inflammation

## Abstract

Endometriosis (EMS) is a long-term inflammatory disease. It represents one of the most prevalent gynecological conditions, impacting an estimated 5% of reproductive women. Therefore, endometriosis contributes to substantial worldwide health challenges and healthcare costs. In EMS disease, endometrial glandular and stromal tissues are abnormally located outside the uterus. Similarly to the natural endometrium, these tissues grow and proliferate in response to estrogen-dependent signals. The pain and limited effectiveness of treatments are often linked to the inflammatory reaction triggered by EMS-associated ectopic tissue. This is especially amplified during the peaks of estrogen release that occur as the menstrual cycle transitions from the proliferative phase to ovulation. Maintaining the integrity of the mucosal lining, defending against pathogenic insults, and controlling physiological processes are all made possible by a healthy, balanced state of gut biomass. Additionally, numerous intestinal bacteria have been discovered to possess estrogen-metabolizing enzymes, which affect the estrobolome and, consequently, influence estrogen-related disorders. Therefore, there is increasing interest in understanding the role of microbiota and the estrobolome in endometriosis pathogenesis. This review will focus on the role of microbiota and the impact of estrobolome alterations in endometriosis pathogenesis.

## 1. Introduction

Endometriosis (EMS) is a long-term and often complicated condition where tissue similar to the lining inside the uterus grows outside of it. This abnormal tissue growth triggers local and systemic inflammation. The precise mechanisms by which endometriosis affects the body is still not fully understood, and it is believed to result from a combination of factors, including genetics, immunity, hormones, and the environment [1]. However, different mechanisms have been suggested to explain how the condition starts and progresses. Gaining a deeper understanding of how endometriosis works is essential for developing better treatment and management options. EMS mainly involves organs in the pelvic area, such as the ovaries, front and back spaces behind the uterus (cul-de-sacs), broad and uterosacral ligaments, uterus itself, fallopian tubes, sigmoid colon, and appendix. In some cases, it can also spread to areas beyond the pelvis, like the diaphragm [2]. Endometriosis impacts around 10% of women during their reproductive years [3] and mostly occurs in younger women (usually 25–45 years) since the growth of endometrial tissue outside the uterus relies on hormone activity from the ovaries. Even though the condition is fairly common, it is often challenging to diagnose, with an average delay of up to 10 years from the symptoms’ onset to receiving an adequate diagnosis [4,5].

There are some hypothetical models to explain how endometriosis can develop. The most well-known is the retrograde menstruation model, which proposes that menstrual blood flows backward through the fallopian tubes, carrying endometrial cells to areas outside the uterus. However, this theory does not fully account for cases in women that do not menstruate [5].

Another model is that endometrial cells spread through the blood vessels or lymphatic system, allowing the disease to affect distant parts of the body. The coelomic metaplasia theory suggests that certain cells in the abdominal lining can change into endometrial-like cells. In addition, genetic and epigenetic factors play a role in increasing risk and influencing EMS progression [1].

Besides gynecological symptoms, up to 90% of EMS patients also report experiencing gastrointestinal issues [6,7]. Recently, growing insight into the role of microbiota and immune system imbalance in various diseases has also highlighted their potential involvement in the development of endometriosis [8]. Researchers have discovered that there is a two-way relationship between the human gut microbiota (GM) and the development of endometriosis, meaning each can influence the other [9]. Since the uterus is not a completely sterile environment, menstrual fluid that flows backward may contain bacteria. This could contribute to inflammation and support the formation and growth of endometriotic lesions. In addition, estrogen’s role in the body goes far beyond reproduction, impacting GM function [10,11], influencing changes in blood composition, and regulating the interplay between the immune system and metabolism [12]. The interaction between estrogen levels and the gut microbiome is known as the “estrobolome,” and it is regulated by some genes that influence how estrogen is metabolized and recycled in the body [13]. In this review, we will focus on the estrobolome’s role in the pathogenesis of endometriosis.

## 2. Female Reproductive Tract Microbiota

The female reproductive tract hosts a distinctive microbiota, accounting for approximately 9% of the total microbial population in the human body [14]. This community comprises a wide range of microorganisms, including bacteria, archaea, fungi, viruses, and, to a lesser extent, parasitic protozoa [15]. This ecosystem plays a crucial role in protecting the genital tract, contributing to vaginal homeostasis and the prevention of obstetric complications, such as preterm birth. In addition, it plays a defensive role against urogenital infections, including bacterial vaginosis (BV), fungal infections, and sexually transmitted diseases, such as HIV infection [16].

Numerous studies have characterized the composition and functional dynamics of the vaginal microbiota, leading to the identification of five distinct community state types (CSTs) based on the predominant bacterial taxa [17]. CSTs I, II, III, and V are typically dominated by *Lactobacillus crispatus*, *L. gasseri*, *L. iners*, and *L. jensenii*, respectively, and are generally associated with eubiotic conditions. In contrast, CST IV is defined by a reduced abundance of *Lactobacillus* spp. and a higher prevalence of anaerobic bacteria such as *Gardnerella*, *Prevotella*, *Atopobium*, *Dialister*, and *Sneathia*, which are frequently linked to vaginal dysbiosis [15,18,19] Table 1.

A clear anatomical gradient in microbial density and composition exists along the female reproductive tract. The lower genital tract—comprising the vagina and ectocervix—contains approximately 100 to 10,000 times more bacteria than the upper tract, which includes the endocervix, uterus, fallopian tubes, and ovaries [20,21]. Along this continuum, the relative abundance of *Lactobacillus* species progressively declines [20].

The vaginal microbiota is highly dynamic and shaped by multiple host-related and environmental factors, including age, hormonal status, menstrual cycle, pregnancy, sexual activity, and lifestyle. Notably, approximately 78% of vaginal samples exhibit *Lactobacillus* dominance, underscoring the genus’s pivotal role in maintaining reproductive tract health [22]. The most common colonizers of a healthy vaginal environment include *L. crispatus*, *L. gasseri*, *L. jensenii*, and *L. iners* [23].

A low vaginal pH (~4.0–4.5), primarily maintained by *Lactobacillus*-derived lactic acid, is critical for suppressing pathogenic overgrowth. Conversely, CST IV, typically associated with a pH > 5.0, facilitates colonization by anaerobic bacteria such as *Gardnerella vaginalis*, *Prevotella*, and *Atopobium*, which are otherwise inhibited in more acidic environments [24]. In such conditions, *L. crispatus* and *L. jensenii* exhibit reduced glycolytic activity, while *G. vaginalis* maintains stable growth and contributes to biofilm formation—particularly at pH levels between 5.5 and 6.5—thereby enhancing infection persistence, antimicrobial resistance, and the neutralization of host defenses [25,26].

This acidification process depends on the metabolic activity of *Lactobacillus* species, which ferment glycogen-derived substrates. Estrogen plays a central role by stimulating glycogen synthesis within the vaginal epithelium. Glycogen is subsequently released into the lumen and enzymatically degraded (e.g., via α-amylase) into glucose and maltodextrins, which are readily utilized by *Lactobacillus* spp. to produce lactic acid [26,27]. The glycolytic efficiency of *L. crispatus* and *L. jensenii* increases under mildly acidic conditions, suggesting a regulatory feedback loop involving hormonal milieu, substrate availability, and microbial selection [26]. This mechanism underlies the maintenance of a protective vaginal ecosystem that limits colonization by pathogens such as *G. vaginalis*, *Chlamydia trachomatis*, *Neisseria gonorrhoeae*, and *Trichomonas vaginalis* [24,28].

Estrogen deficiency, such as during menopause, decreases epithelial glycogen levels, leading to diminished *Lactobacillus* abundance, elevated pH, and the overgrowth of anaerobic bacteria such as *Gardnerella*, *Mycoplasma*, and *Prevotella* [16]. Ethnic disparities have also been observed: women of Caucasian and Asian descent generally display higher *Lactobacillus* abundance compared to women of African or Hispanic ancestry [29]. Topical estradiol therapy has demonstrated efficacy in restoring *Lactobacillus* dominance and lowering vaginal pH in postmenopausal women [30].

Shifts in microbial composition—whether physiological or pathological—can influence fertility, embryo implantation, and outcomes of assisted reproductive technologies [20]. Although *L. iners* is frequently dominant, it is considered metabolically and functionally unstable and is often associated with transitions toward dysbiosis [31]. Bacterial vaginosis (BV), a prevalent dysbiotic condition, is marked by *Lactobacillus* depletion and anaerobic overgrowth—most notably of *G. vaginalis*—which contributes to biofilm formation and high recurrence rates [32,33]. CST IV-B has also been associated with genitourinary syndrome of menopause (GSM) and related symptoms such as leukorrhea and burning.

Reduced *Lactobacillus* abundance has been correlated with increased severity of vulvovaginal symptoms, while specific genera—such as *Escherichia-Shigella*, *Anaerococcus*, *Finegoldia*, *Peptoniphilus*, *Enterococcus*, and *Streptococcus*—have been implicated in symptomatology [34]. However, the interplay between circulating estrogen, glycogen availability, and microbial composition remains incompletely understood. For instance, a cross-sectional study in postmenopausal women found an association between elevated estrone levels and *Lactobacillus* dominance, while glycogen levels did not correlate with symptom severity—suggesting that additional systemic or local modulators are involved [35].

Other host factors, including obesity, genetic background, hormonal contraceptive use, and surgical history (e.g., hysterectomy), may influence the risk of vaginal dysbiosis [29,36]. In this context, *Lactobacillus*-based probiotics are emerging as promising therapeutic options to restore vaginal eubiosis, particularly in cases of recurrent infections such as pregnancy-associated vulvovaginal candidiasis [37]. Finally, mounting evidence confirms that the upper reproductive tract harbors distinct microbial communities that likely play important roles in local immune modulation and reproductive outcomes [21].

## 3. Gut Microbiota and the Estrobolome

### 3.1. Introduction to the Estrobolome

The gut microbiota is now recognized as a real “endocrine organ” due to its ability to produce bioactive metabolites that, once absorbed into the systemic circulation, can modulate distant physiological functions, including host hormonal balance [38].

According to Baker et al. [11] E, the interaction between estrogens and the gut microbiota constitutes a true “estrogen–microbiome axis”, which plays a crucial role in the regulation of female hormonal homeostasis.

Recent scientific evidence suggests that GM homeostasis plays a pivotal role in women’s health, with an emerging association between intestinal dysbiosis and various gynecological and endocrine disorders, including endometriosis, polycystic ovary syndrome, and hormone-dependent cancers such as breast, cervical, and ovarian carcinomas. However, the molecular mechanisms underlying these associations remain largely unclear. One of the main models proposed involves interaction between the gut microbiota and estrogen metabolism. In this context, the term *estrobolome* has been coined to describe the collection of bacterial genes involved in modulating the enterohepatic estrogens’ circulation, thereby affecting their systemic bioavailability [39].

### 3.2. Mechanism: β-Glucuronidase Activity

Endogenous estrogens, primarily synthesized in the ovaries, adrenal glands, and adipose tissue, are inactivated in the liver through conjugation processes (mainly glucuronidation) and subsequently excreted via bile and feces [40]. Beta-glucuronidase and beta-glucosidase are key enzymes produced by gut bacteria involved in estrogen’s conjugation and reabsorption into the bloodstream. The gut microbiota’s composition directly impacts circulating estrogen levels. Obese patients present an overexpression of this mechanism, mainly due to peripheral aromatization of testosterone and androstenedione to estradiol and estrone [41]. As early as the 1960s, estrogens were detected in human feces, suggesting a direct GM role in regulating their systemic reabsorption through the deconjugation of estrogen conjugates [42]. This process is mainly mediated by β-glucuronidase enzyme, produced by specific intestinal bacteria, which can hydrolyze conjugated estrogens, rendering them active and available for reabsorption [43].

At the molecular level, a recent study identified 35 distinct variants of bacterial β-glucuronidases (GUSs), of which 17 demonstrated the ability to deconjugate estrogen metabolites, specifically estrone-3-glucuronide and estradiol-17-glucuronide [43]. These GUS enzymes were structurally categorized into five subgroups—Loop 1, mini-Loop 1, Loop 2, mini-Loop 2, and FMN-binding—based on active site architecture. Among these, Loop 1-type enzymes were found to be the most efficient in catalyzing estrogen deconjugation.

Mechanistically, GUS-catalyzed hydrolysis occurs in two steps: (1) the formation of a covalent enzyme–substrate intermediate; (2) the subsequent release of the active estrogen. The process involves a deep active-site pocket where key residues—such as glutamate and aspartate—facilitate the cleavage of the glucuronide bond. X-ray crystallographic studies revealed a high degree of hydrogen bonding and steric complementarity between Loop 1 GUS enzymes and their estrogen substrates, suggesting substrate specificity at the molecular level.

Notably, the targeted inhibition of Loop 1 GUS activity using small molecules such as UNC10201652 has been shown to significantly reduce estrogen reactivation in ex vivo human fecal samples, underscoring its potential therapeutic relevance.

The diversity of bacterial β-glucuronidases has recently been characterized, with six major enzymatic groups identified [44]. Key β-glucuronidase-producing bacteria belong to the phyla *Bacteroidetes* and *Firmicutes*, including *Bacteroides fragilis*, *Bacteroides thetaiotaomicron*, and *Clostridium perfringens*.

### 3.3. Influence on Estrogen-Dependent Diseases

The enzymatic activity of these microbes contributes to increasing free estrogen levels in systemic circulation, with potential implications in the development of estrogen-dependent diseases, mainly breast cancer [45].

As a result, alterations in GM composition and function, triggered by environmental factors such as diet, medications, or antibiotics, may significantly influence estrobolome activity. This, in turn, can modulate not only estrogen bioavailability but also the risk of developing hormone-dependent conditions, including breast and endometrial cancer, and, of course, endometriosis [40] (Figure 1).

### 3.4. Systemic—Local Microbiota Axis

The metabolic activity of the gut microbiota functions within a broader systemic–local axis, whereby intestinal microbes modulate circulating estrogen levels—primarily via β-glucuronidase activity—and, in turn, influence peripheral estrogen-responsive tissues such as the vaginal mucosa, endometrium, and mammary glands through a combination of microbial metabolites, hormonal signaling, and immune modulation [11].

Disruption of the estrobolome’s equilibrium may lead to elevated systemic levels of free estrogens and concurrently alter the local microenvironment, fostering inflammation, cellular proliferation, and immune dysregulation—hallmark processes implicated in the pathogenesis of various estrogen-dependent disorders.

These mechanisms are schematically shown in Figure 2A,B, which depict, respectively, the role of the gut microbiota in intestinal dysbiosis and immune activation, and the molecular cascade triggered by the interaction between estrogens and their receptors in peripheral tissues.

In summary, a functional axis can be hypothesized between the gut microbiota and female reproductive disorders, mediated by the estrobolome’s ability to regulate the enterohepatic circulation of estrogens and, consequently, the serum concentration of their bioactive forms. These hormones may exert local effects—such as promoting Lactobacillus dominance in the genital microbiota—or act systemically through their interaction with estrogen receptors in target tissues. This mechanism may contribute to the development or modulation of estrogen-dependent conditions, including endometriosis and breast cancer (Table 2).

We summarized the dominant microbial taxa identified in the vagina, endocervix, endometrium, fallopian tubes, and ovary (including follicular fluid). In addition, we outlined key modulators of microbial composition (such as hormonal, anatomical, and pathological factors), discussing their known or potential roles in reproductive health and disease, with a focus on gynecological conditions including endometriosis and PCOS.

## 4. Estrobolome Dysregulation in Endometriosis

It is believed that a balanced gut microbiome helps maintain stable levels of circulating estrogen, while its dysbiosis can disrupt this balance and potentially lead to estrogen-related diseases. There is evidence suggesting that the estrobolome plays a role in estrogen-related cancers such as colon and breast cancer [40,48] with increased β-glucuronidase activity in the gut, which is linked to both pathological conditions. Since endometriosis is related to estrogen, variations in the estrobolome might also have a role in increasing the likelihood of developing this condition [43]. As previously mentioned, estrogens are crucial for the adequate development and functioning of the female reproductive system, but they play a relevant role in EMS onset and progression [49,50]. Estrogen and phytoestrogens bind to estrogen receptors, which are then taken into cells and act as transcription factors, driving the expression of genes that regulate cell growth. In healthy conditions, the gut bacteria support the good development, turnover, and function of host epithelial cells [51]. In addition, they regulate mucus thickness, the function of the transepithelial barrier, and immune responses, all of which are essential features of a healthy epithelium [51,52,53]. Bacteria can break down the protective mucus layer in the intestine and directly interact with enterocytes, often leading to increased local and systemic inflammation. Additionally, dysbiosis can cause immune dysfunction and disrupt intestinal barrier function, allowing the gut microbiota and bacterial components to migrate beyond the gut [51,52,53]. Finally, the GM affects the composition and function of mucosal T-cells (such as Th1, Th17, Treg), and so its dysbiosis disrupts the balance of these immune cells, triggering inflammation and disease. Elevated levels of systemic inflammation are EMS-linked, suggesting that the gut microbiota might play a role in this disorder by contributing to or disrupting an inflammatory feedback loop [54]. Pain symptoms in endometriosis are likely connected to both systemic and localized inflammation [55]. As previously described, the gut microbiota, mainly when altered, plays a role in reactivating estrogens, in particular through the production, as previously reported, of the enzyme β-glucuronidase that breaks down glucuronic acid conjugates [43,56], increasing the levels of active estrogens, which in turn leads to the amplified activation of target cells responsive to the beta estrogen receptor (ERβ) [57,58]. The activity of β-glucuronidase is crucial for the production of potentially harmful and cancer-causing metabolites in the gut, as well as for the reabsorption of various substances, including estrogens; this process promotes an excess of estrogens, which fuels endometriosis’ development. It also offers a potential explanation for how GM alterations could contribute to the disease’s onset and progression. Estrogens that remain active in the gut can be carried, through the bloodstream, to distant mucosal tissues, like the endometrium. This explains how the gut microbiota helps maintain estrogen balance not only in the intestines but also in other host districts. Altered Firmicutes to Bacteroidetes ratios, documented in EMS women, may influence estrogen metabolism, as bacteria from these groups carry genes related to glucuronidase activity [59]. A recent study by Pérez-Prieto et al. investigated the connection between estrogen metabolism and endometriosis by comparing microbial enzyme activity related to the estrobolome in EMS women and controls; however, estrobolome-associated enzymes’ sequence reads were not significantly different between the groups [60]. Another recent study involving women of reproductive age used their stool and urine samples to examine enzyme activity, microbial composition, and metabolic profiles, aiming to determine whether EMS patients have distinct gut microbiota or changes in estrogen metabolism. Although there were no major differences in gut β-glucuronidase activity, microbial diversity, or overall bacterial abundance between the two groups, women with endometriosis showed a higher presence of bacteria from the *Erysipelotrichia* class in their fecal samples, along with elevated levels of four estrogen-related compounds [61]. While studies on the fine interactions between the estrobolome and the microbiome in endometriosis are still scarce, there are several potential mechanisms through which they could interact. This is relevant not just for host estrogen metabolism but also for the onset of inflammation or the production of secondary metabolites. A GM imbalance can also lead to shifts in the metabolome, potentially resulting in elevated levels of neuroactive compounds like serotonin, glutamate, short-chain fatty acids (SCFAs), and gamma-aminobutyric acid (GABA) [62,63]. For instance, specific bacteria species can produce SCFAs by fermenting dietary fibers. SCFAs can regulate estrogen metabolism by affecting the expression of enzymes that process estrogen in the liver and other tissues. These metabolites can reach the brain and activate neural receptors, including those on GnRH neurons, which in turn trigger a chain of hormonal signals that increase estrogen production by the ovaries. Finally, changes in estrogen levels and signaling in endometriosis, driven by estrobolome alterations, the increased activity of estrogen-producing enzymes, and the irregular expression of estrogen receptors, may also influence the composition of the genital microbiota [64].

## 5. Future Perspectives

Imbalances in the gut and reproductive tract microbiomes can interfere with normal immune responses, triggering inflammation by increasing pro-inflammatory cytokines and altering immune cells’ function. This ongoing immune imbalance can lead to chronic inflammation, which supports processes like tissue adhesion and new blood vessel formation, key factors in the development and progression of endometriosis. Recent research shows that endometriosis can alter the microbiota, and that antibiotics may be a potential treatment. As a developing field of study, the connection between the microbiota and endometriosis remains poorly understood. Much of the existing data is based on small sample sizes and lacks strong, randomized controlled trials, making it difficult to draw appropriate conclusions. The GM influence on estrogen regulation holds promising potential for innovative treatments. However, randomized controlled trials are needed to more clearly define how effective such therapies could be in clinical conditions influenced by estrogens. Analysis of the gut microbiome could be further advanced through metagenomic techniques, allowing researchers to measure the presence of genes responsible for producing β-glucuronidase and other genes involved in estrogen metabolism. Much more research is needed, especially to define the “core” microbiome and clarify the mechanisms behind the link between the microbiota and endometriosis.

## Figures and Tables

**Figure 1 cimb-47-00502-f001:**
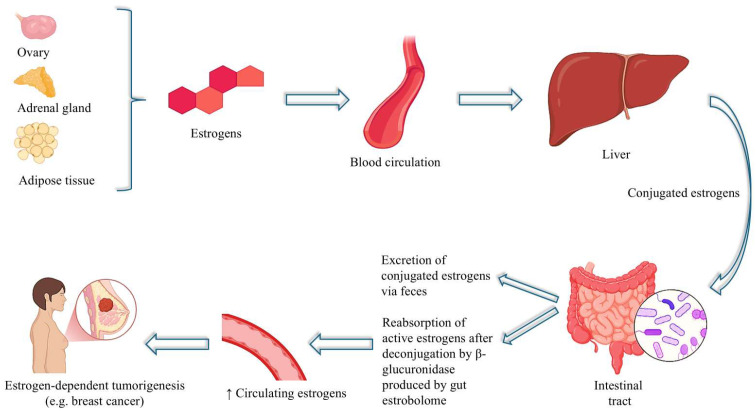
A schematic representation of the enterohepatic circulation of estrogens and the role of the gut microbiota (estrobolome) in modulating systemic estrogen levels.

**Figure 2 cimb-47-00502-f002:**
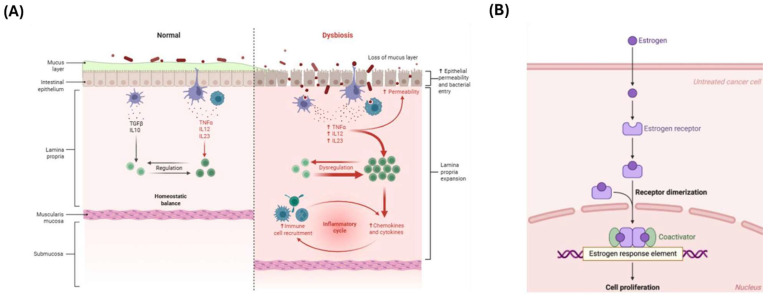
(**A**) **Intestinal dysbiosis**. The disruption of mucosal integrity and increased epithelial permeability facilitate immune activation, triggering the release of pro-inflammatory cytokines such as TNF-α, IL-12, and IL-23 and promoting immune cell infiltration. This initiates a self-perpetuating cycle of chronic intestinal inflammation [https://doi.org/10.1056/NEJMra0804647, access date 1 May 2025]. (**B**) **Estrogen signaling in peripheral tissues**. Estrogens bind to intracellular nuclear receptors, inducing receptor dimerization and the recruitment of transcriptional complexes to estrogen response elements (EREs) within target gene promoters. This molecular cascade regulates key processes including cell proliferation, differentiation, and immune modulation [doi:10.1007/s10549-019-05154-7, access date 1 May 2025].

**Table 1 cimb-47-00502-t001:** Vaginal community state types (CSTs) and their association with microbial composition, hormone levels, pH, and anatomical localization. While CSTs I–III and V reflect healthy, estrogen-rich, acidic environments, CST IV is associated with hypoestrogenism, elevated vaginal pH, and dysbiosis.

CST	Dominant Anatomical Site	Dominant Species	Mean pH	Estrogen Levels	Epithelial Glycogen
I	Lower vagina	*L. crispatus*	~4.0	High	High
II	Mid/upper vagina	*L. gasseri*	~4.2	Moderate	Moderate
III	Upper vagina/cervix	*L. iners*	~4.4	Variable	Variable
IV	Cervix, endometrium	Mixed anaerobes	>5.0	Low	Low
V	Vagina (ascending)	*L. jensenii*	~4.2	Moderate	Moderate

**Table 2 cimb-47-00502-t002:** Summary of microbial composition, influencing factors, and clinical implications across different compartments of female reproductive tract.

Compartment	Dominant Microbial Composition	Influencing Factors	Role and Clinical Implications	References
Vagina	Prevalence of *Lactobacillus* spp. (CST I–III, V); CST IV enriched with anaerobes such as *Gardnerella* and *Atopobium*	Estrogen levels, menstrual cycle, age, ethnicity, pregnancy, local estradiol therapy, antibiotic use, contraception, and menopause	Lactobacillus-dominated microbiota supports protective vaginal environment; microbial imbalance associated with infections, menopause-related symptoms, and reduced fertility	[16,18,22,29,30]
Endocervix	*Lactobacillus iners*, *Lactobacillus crispatus, Prevotella*, *Sphingobium*, *Propionibacterium acnes* and *Pseudomonas*	Menstrual cycle phase, parity (nulliparous vs. multiparous), use of medications (e.g., herbal treatments), and gynecological conditions such as adenomyosis and endometriosis	Maintains protective barrier against pathogens through acidification; microbial imbalance may increase infection risk, impair fertility, and is associated with gynecological disorders	[20]
Endometrium	*Lactobacillus* spp., *Gardnerella*, *Atopobium*, *Streptococcus*, *Bifidobacterium*, and *Prevotella*	Estrogen and progesterone levels, intrauterine interventions (e.g., embryo transfer, biopsies), assisted reproductive technologies (ART), local immune and inflammatory status, and gynecological conditions (e.g., endometriosis, cancer)	Balanced endometrial microbiota may support embryo implantation; dysbiosis has been linked to inflammation, implantation failure, and endometrial disorders	[14,46]
Fallopian tubes	*Bacteroides*, *Corynebacterium*, *Lactobacillus*, *Coprococcus*, and *Hymenobacter*	Shaped by ascending microbial migration, interindividual variability, and anatomical location	Microbial presence detected in healthy fallopian tubes; potential physiological role still under investigation	[20]
Ovary/Follicular fluid	*Staphylococcus aureus*, *Streptococcus* spp., *Enterococcus* spp., *Lactobacillus* spp. and *Candida albicans*	Underlying gynecological conditions (e.g., endometriosis, PCOS), history of genital tract infections, and invasive gynecological procedures	Microorganisms in follicular fluid associated with reduced fertilization rates, especially in patients with endometriosis or PCOS; possible implications for pregnancy outcomes remain inconclusive	[47]

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
