# Peer review of "Unraveling the Contribution of Estrobolome Alterations to Endometriosis Pathogenesis"

_cimb, 2025, doi:10.3390/cimb47070502_

Round 1
Reviewer 1 Report
Comments and Suggestions for Authors
The manuscript in case provides a comprehensive overview of the complex interplay between the gut microbiota, specifically the estrobolome, and endometriosis. It clearly identifies a significant gap in understanding the precise mechanisms through which gut microbiota alterations influence endometriosis, highlighting potential therapeutic targets. The manuscript brings forward the emerging field of microbiota involvement in estrogen metabolism and its implications in endometriosis pathogenesis, a relatively novel and promising area of research.
- Frequent paragraphs that need rephrasing for better understanding, such as: line 21, line 32 “The right way”
- Line 46-50: provide adequate reference
- Line 175: after “….and feces” you may add “Beta-glucuronidase and beta-glucosidase are key enzymes produced by gut bacteria involved in estrogens conjugation and reabsorbtion back into the bloodstream. The gut microbiota’s composition directly impacts circulating estrogen levels. Obese patients present an overexpression of this mechanism, mainly due to peripheral aromatization of testosterone and androstenedione to estradiol and estrone.” You may cite PMID 36900020 together with [36]
- While Figure 1 and Table 1 are helpful, additional figures illustrating the pathways of microbiota-induced inflammation and estrogen regulation could significantly improve the visual appeal and understanding of complex mechanisms described
Syntax errors. Manuscript would benefit from the lecture of a native English speaker
Reviewer 2 Report
Comments and Suggestions for Authors
Dear Authors,
Please provide your responses to the following comments/suggestions:
- Line 32: “The right way by endometriosis affects the body…” What does it mean? It’s not clear what the author wants to mention. Please revise this sentence accordingly.
- In the introduction section the paragraph transitions are not linked, especially between the mechanisms of ED and the microbiome section. Please revise this and write a connecting line before the microbiome section starts.
- The acronym “ED” for endometriosis can be confused with “erectile dysfunction.” It’s better to use “EMS” or write “endometriosis.” In the manuscript.
- Please expand the writing about how the estrobolome modulates estrogen metabolism and the role of β-glucuronidase-producing bacteria in estrogen reactivation in the introduction section.
- There are repetitive references to the role of estrogen in regulating Lactobacillus and glycogen metabolism appear in multiple places, it will be better to combine this section in a single paragraph in a descriptive way, without any repetition.
- Please include a figure or schematic diagram summarizing CSTs, hormonal impact, or microbial progression from lower to upper tract for better understanding.
- The authors mention that interactions exist (e.g., between estrogen and microbiota) but did not mention the mechanistic details (e.g., how Lactobacillus metabolizes glycogen, or how estrogen levels are sensed by microbial communities). Please include these details.
- The CST-specific pH data are mentioned, but please explain why higher pH (>5.0) in CST IV is problematic (supports anaerobe overgrowth, biofilm formation, pathogen survival).
- The section Gut microbiota and the estrobolome: needs to be divided in to subsections: like Introduction to the Estrobolome, mechanism: β-glucuronidase activity, influence on Estrogen-dependent diseases and systemic–local microbiome axis for better understanding.
Round 2
Reviewer 1 Report
Comments and Suggestions for Authors
The authors have provided a thoroughly revised version. Good work1
Reviewer 2 Report
Comments and Suggestions for Authors
Dear Authors,
Thank you for revising the manuscript as per comments/suggestions.
All the best.